# Association between Intraoperative Blood Transfusion, Regional Anesthesia and Outcome after Pediatric Tumor Surgery for Nephroblastoma

**DOI:** 10.3390/cancers14225585

**Published:** 2022-11-14

**Authors:** Sarah D. Müller, Christian P. Both, Christoph Sponholz, Maria Theresa Voelker, Holger Christiansen, Felix Niggli, Achim Schmitz, Markus Weiss, Jörg Thomas, Sebastian N. Stehr, Tobias Piegeler

**Affiliations:** 1Department of Anesthesiology and Intensive Care, University Hospital Leipzig, 04275 Leipzig, Germany; 2Department of Anesthesia, University Children’s Hospital Zurich, 8032 Zurich, Switzerland; 3Department of Anesthesiology and Intensive Care Medicine, University Hospital Jena, 07740 Jena, Germany; 4Department of Pediatric Oncology, Hematology and Hemostaseology, University Hospital Leipzig, 04103 Leipzig, Germany; 5Department of Pediatric Oncology, University Children’s Hospital Zurich, 8032 Zurich, Switzerland; 6EuroPeriscope: The ESA-IC Onco-Anaesthesiology Research Group

**Keywords:** anesthesia, cancer, blood products, nephroblastoma, outcome

## Abstract

**Simple Summary:**

Previously published data suggest anesthesiologic interventions during the perioperative period might play a major role in determining patient outcome after tumor surgery. However, only scarce data are available regarding this question in pediatric patients. This retrospective, multicenter study aimed to assess such a potential effect in pediatric patients undergoing cancer surgery for nephroblastoma. Data from 65 patients were analyzed. Intraoperative administration of erythrocyte concentrates was associated with a reduction in recurrence-free survival. The use of regional anesthesia or the choice of anesthetic had no effect. However, regional anesthesia was associated with fewer ICU transfers, a shortened hospital stay and a decreased postoperative neutrophil-to-lymphocyte ratio. The current study provides the first evidence for a possible association between anesthesia and outcome after pediatric cancer surgery. Children undergoing tumor surgery might therefore benefit from an optimized anesthetic regimen—including the use of regional anesthesia—and more restrictive blood transfusion management.

**Abstract:**

Background: Recent data suggest that anesthesiologic interventions—e.g., the choice of the anesthetic regimen or the administration of blood products—might play a major role in determining outcome after tumor surgery. In contrast to adult patients, only limited data are available regarding the potential association of anesthesia and outcome in pediatric cancer patients. Methods: A retrospective multicenter study assessing data from pediatric patients (0–18 years of age) undergoing surgery for nephroblastoma between 2004 and 2018 was conducted at three academic centers in Europe. Overall and recurrence-free survival were the primary outcomes of the study and were evaluated for a potential impact of intraoperative administration of erythrocyte concentrates, the use of regional anesthesia and the choice of the anesthetic regimen. The length of stay on the intensive care unit, the time to hospital discharge after surgery and blood neutrophil-to-lymphocyte ratio were defined as secondary outcomes. Results: In total, data from 65 patients were analyzed. Intraoperative administration of erythrocyte concentrates was associated with a reduction in recurrence-free survival (hazard ratio (HR) 7.59, 95% confidence interval (CI) 1.36–42.2, *p* = 0.004), whereas overall survival (HR 5.37, 95% CI 0.42–68.4, *p* = 0.124) was not affected. The use of regional anesthesia and the choice of anesthetic used for maintenance of anesthesia did not demonstrate an effect on the primary outcomes. It was, however, associated with fewer ICU transfers, a shortened time to discharge and a decreased postoperative neutrophil-to-lymphocyte ratio. Conclusions: The current study provides the first evidence for a possible association between blood transfusion as well as anesthesiologic interventions and outcome after pediatric cancer surgery.

## 1. Introduction

The incidence of nephroblastoma, the most frequent solid malignant tumor in children, is also often referred to as Wilms’ tumor (WT) and peaks between age two and four [1]. The establishment of a multimodal therapy regimen led to a significant improvement of outcome, and so overall survival is currently expected to be over 90% [2]. However, cases with relapse still have a poor prognosis [3,4]. As a consequence, research has especially focused on risk minimization for cancer recurrence [5]. 

During surgery, tumor cells are shed into the bloodstream and the lymphatic system [6]. These circulating tumor cells (CTCs) can form new metastatic sites, even after complete tumor resection [7]. Surgical trauma may further enhance the pathogenesis of metastasis by initiating several complex immunomodulatory mechanisms, which lead to a greater chance for CTCs to escape immunosurveillance [6,8,9]. As these processes might be affected by the choice of the anesthetic regimen, the perioperative period might be crucial regarding long-term survival [10]. Several recent retrospective analyses revealed that regional anesthesia and local anesthetics might have beneficial effects on patients’ outcomes after cancer surgery [11,12] based on the strong anti-inflammatory properties of the drugs [13,14,15]. However, only scarce data are available regarding a possible impact of anesthesia on outcome after cancer surgery in children.

Anemia has also been identified as an independent prognostic factor for the survival of cancer patients [16]. However, transfusion-related immunomodulation might lead to an enhancement of inflammatory mechanisms, thus favoring cancer recurrence [17]. Again, little is known regarding these potential interactions in the pediatric setting.

This current retrospective multicenter study therefore aimed to assess potential effects of regional anesthesia and/or intraoperative blood transfusions on outcome—i.e., overall and recurrence-free survival—in pediatric patients undergoing cancer surgery for nephroblastoma. Additionally, the impact of both interventions on the length of stay in the intensive care unit (ICU), the time to hospital discharge as well as on the neutrophil-to-lymphocyte ratio (NLR) were evaluated.

## 2. Methods

### 2.1. Study Design

This retrospective cohort study was conducted at the following three academic centers after receiving approval from the respective local ethics committee: University Hospitals Leipzig (protocol number: 238/19-ek), and Jena (2019-1567-Daten), Germany, and University Children’s Hospital Zurich, Switzerland (2019-01250). Electronic medical records were screened for patients between age 0 and 18 years undergoing surgery for nephroblastoma between 2004 and 2018. Incomplete records, records blocked for research and those from patients only receiving sedation were excluded from analysis.

### 2.2. Participants and Data Collection

Data were extracted from the centers’ anesthesia patient data management systems (Leipzig: Copra 5, Version 5.24.797, Jena: Copra 5, Version 5.24.974, Copra System GmbH, Berlin, Germany; Zurich: MetaVision, Version 5.46.44, iMDsoft, Duesseldorf, Germany) as well as from the individual patients’ electronic medical records.

The primary outcome of the study was the impact of intraoperative erythrocyte transfusion and the use of regional anesthesia on overall survival (OS) and recurrence-free survival (RFS) after the surgical removal of the tumor.

As secondary outcomes, the influence of the use of regional anesthesia and the choice of the anesthetic for maintenance of anesthesia (balanced anesthesia including volatile anesthetics vs. total intravenous anesthesia (TIVA) with propofol) on OS and RFS was examined. The length of stay on the intensive care unit (ICU) and the time to discharge to home after surgery were noted as well. Additionally, the neutrophil-to-lymphocyte ratio (neutrophil count/lymphocyte count, NLR) at three different timepoints (at the time of diagnosis, pre-operative, post-operative (data ranging from post-operative day 1 to a maximum of post-operative day 8 depending on the available data)) was recorded.

Besides epidemiological data, the duration of anesthesia and surgery as well as the type of drugs used for maintenance of anesthesia, the amount and the type of intraoperatively administered fluids were also recorded.

### 2.3. Statistical Analysis

Categorical data are given as the number of patients along with their corresponding percentage of their total group. Differences between proportions of qualitative data were assessed using χ^2^ or Fisher’s Exact test. Quantitative data were assessed for normal distribution using the Shapiro-Wilk test. Normally distributed data are expressed as mean (±standard deviation, SD), and Student’s *t*-test was used for the comparisons between groups. Non-parametric data are shown as median with interquartile range (IQR) and were compared using a Mann-Whitney U test.

For the analysis of OS and RFS—defined as the timespan reaching from the day of initial surgery to the death of the patient or the date of the diagnosis of relapse, respectively—a Kaplan-Meier curve was plotted and analyzed using a log-rank test. Patients who did not reach the designated follow-up period of five years (observation period until 31 December 2020) were marked as censored. Besides the *p*-values of the result of the log-rank test, hazard ratios (HR) with their respective confidence intervals (CI) are reported. As the log-rank test regarding the RFS and PRBC transfusion reached statistical significance, an additional multivariate Cox regression analysis of the RFS was conducted. As the two groups (PRBC vs. No PRBC) differed regarding the distribution of the different stages of the disease, the Cox regression aimed at correcting for this matter by including the stage of the disease along with the intraoperative use of PRBC in the analysis. In accordance with previous reports and due to the fact that survival rates do not differ significantly between certain stages, patients were stratified into a first group containing stages I and II or into a second group including stages III, IV and V for the Cox regression [18].

Neutrophil-to-lymphocyte ratio data were assessed using two-way analysis of variance (ANOVA) with corresponding treatments and different timepoints as factors to be tested. From these analyses, the F ratio with the degrees of freedom for each of these factors and for the interaction between them are reported as necessary. Bonferroni post hoc testing was conducted to control for a family-wise error rate of <0.05.

A *p*-value < 0.05 was considered to be statistically significant. Analyses were performed using GraphPad Prism software, Version 9 (GraphPad Software, San Diego, CA, USA) or SPSS software (Version 27, IBM Corp, Armonk, NY, USA).

## 3. Results

### 3.1. Baseline and Perioperative Data

In total, electronic records from 66 patients (Leipzig: 16, Jena: 15, Zurich: 35) were assessed. One patient had to be excluded due to incomplete documentation. Baseline characteristics of all patients are summarized in Table 1.

Regarding demographic data, no differences could be detected between patients who received packed red blood cells (PRBC group, *n* = 18) or did not (No PRBC group, *n* = 47) intraoperatively. However, the two groups differed in terms of the initial stage (*p* = 0.005), with a tendency for a larger number of patients in stages II (22% vs. 12%) and III (39% vs. 9%) in the PRBC group (all Table 2).

The duration of surgery as well as of anesthesia was longer in the PRBC group (surgery: median 271 (IQR 106) vs. 171 (66) minutes, *p* = 0.001; anesthesia: 376 (123) vs. 278 (76) minutes, *p* < 0.001). Patients in the PRBC group also received higher amounts of crystalloids (80 (47) vs. 67 (41) mL/kg body weight (BW), *p* = 0.005), although the amount in relation to body weight over time was comparable (13 (9) vs. 14 (6) mL/kg BW/h, *p* = 0.40, all Table 2).

Regional anesthesia (RA) was used in 45 out of 65 patients (69%). Of these, 10 patients received a caudal anesthesia (single-shot), and 35 patients were provided with an epidural catheter (Table 1). Table 3 shows baseline characteristics as well as perioperative data for patients with or without RA as part of the anesthetic regimen (RA group vs. No RA group). Again, a difference regarding the initial tumor staging could be detected (*p* = 0.029), with higher stages and lower ASA physical status classification scores being present in the RA group (RA: 69% ASA II and 31% ASA III, no RA: 40% ASA II and 55% ASA III, *p* = 0.025) as well as a significantly higher amount of colloids administered to patients with RA (23 (16) vs. 10 (24) mL/kg BW/h, *p* = 0.04).

Additionally, the use of RA was associated with a significantly lower rate of postoperative ICU admissions (31% vs. 75%, *p* = 0.001, all Table 3).

### 3.2. Primary Outcome: Influence of Intraoperative Blood Transfusions on Overall and Recurrence-Free Survival

Overall, three children died and seven children suffered from a relapse during the observation period. Intraoperative administration of PRBCs had no impact on OS (log-rank test: HR 5.37, 95% CI 0.42–68.4, *p* = 0.124, Figure 1A and Table 4). However, RFS was significantly reduced in patients receiving PRBCs intraoperatively (log-rank test: HR 7.59, 95% CI 1.36–42.2, *p* = 0.004, Figure 1B and Table 4).

The additional Cox regression analysis of the RFS regarding the effect of PRBC transfusion correcting for the stage of the disease still revealed a statistically significant effect of intraoperative blood transfusion (HR 5.99, 95%CI 1.12–31.96, *p* = 0.04), while the initial stage did not have such an effect (HR 2.07, 95%CI 0.45–9.53, *p* = 0.351, all Table 5).

### 3.3. Secondary Outcomes

#### 3.3.1. Influence of Regional Anesthesia and Maintenance of Anesthesia on Overall and Recurrence-Free Survival

Neither the perioperative use of RA nor the type of anesthetic used for maintenance of anesthesia demonstrated a statistically significant impact on OS or RFS (Figure 2 and Table 4).

#### 3.3.2. Length of Stay on Intensive Care Unit and Time to Hospital Discharge

The length of stay (LOS) on the ICU appeared longer (median 2 (3) vs. 1 (2) days, Figure 3(Ai)) in the PRBC group. The use of RA was also associated with a reduction of the median length of stay by 50% (2 (4) vs. 1 (1) days; Figure 3(Aii)). However, both differences were not found to be statistically significant (PRBC: *p* = 0.36; RA: *p* = 0.17).

Intraoperative PRBC transfusion did not lead to alterations regarding the time to discharge from the hospital after surgery (7 (9.25) vs. 7 (3) days, *p* = 0.55, Figure 3(Bi)). However, the use of RA was associated with a significant reduction of this variable by three days (10 (7.75) vs. 7 (3) days, *p* = 0.01, Figure 3(Bii)).

#### 3.3.3. Neutrophil-to-Lymphocyte Ratio

The NLR was retrospectively assessed at the time of diagnosis, PRE-operative and POST-operative (postoperative day 1 to 8). Not all 65 patients had complete blood counts drawn at these distinct timepoints; therefore, the number of patients to be assessed varied between the different timepoints (*n* = 61 diagnosis, *n* = 56 PRE-operative, *n* = 45 POST-operative). When comparing the PRBC with the No PRBC group regarding NLR (Figure 4A), analysis using a 2-way ANOVA revealed a statistically significant effect of the time (F(2, 156) = 14.58, *p* < 0.001), whereas the administration of PRBCs did not have an effect (F(2, 156) = 1.87, *p* = 0.17, Figure 4A). There was a decrease in the NLR in both groups (PRBC and No PRBC) after neoadjuvant chemotherapy (values at diagnosis vs. PRE-operative) and another increase from PRE- to POST-operative, again in both groups, which did not differ significantly from each other (POST-operative, *p* = 0.82, all Figure 4A).

A comparable pattern with a decrease in both groups from the time of diagnosis until PRE-operative and the subsequent increase after surgery could be observed when comparing the NLR in patients who had received a combination of general anesthesia and RA or who only received general anesthesia (Any RA vs. No RA, Figure 4B). However, the postoperative rise in the NLR was 48% lower in the RA group (mean 4.36 ± 3.36 vs. 2.30 ± 2.09, *p* = 0.001). This was also reflected by the results of the 2-way ANOVA, which did not only reveal a significant impact of the time (F(2, 153) = 21.36, *p* < 0.001), but also of the use of RA (F(1, 153) = 5.805, *p* = 0.017) and the interaction of the terms (F(2, 153) = 4.42, *p* = 0.014, all Figure 4B).

## 4. Discussion

The main finding of this retrospective multicenter study was that intraoperative transfusion of PRBCs reduced RFS in pediatric patients undergoing cancer surgery for nephroblastoma, while it had no effect on overall survival.

This once more underlines the importance of perioperative management for the prevention of cancer recurrence and is—to our knowledge—the first study to provide such data for children. The results of the current study are in accordance with previous data obtained from adults, which have also established a potential association between perioperative blood transfusion and reduced survival in patients with cancer [19,20]. The strong pro-inflammatory and immunomodulatory effect of allogeneic blood products leading to a phenomenon called “transfusion-related immunomodulation” (TRIM) [17]. TRIM might enhance the suppression of the activity of Natural Killer (NK) cells [17], which play an important role in cancer immunosurveillance [21]. A perioperative decrease in NK cell activity by transfusion of allogeneic blood products might therefore favor an escape of CTCs from the surveillance system and promote the formation of new metastatic lesions and cancer recurrence [10,21].

The presented data suggests that a more restrictive blood transfusion regimen might lead to more favorable outcomes and prolonged (recurrence-free) survival rates. In general, pediatric oncology patients tend to have a high utilization of blood products: a recent database analysis from a large academic center in Canada revealed that 66% of all pediatric oncology patients received at least one PRBC or platelet unit during their treatment [22]. In our study, the transfusion rate was 27.7%, which is in accordance with another study evaluating the effects of perioperative blood transfusion in pediatric patients undergoing surgery for solid tumors [23]: here blood transfusions were associated with more postoperative complications, an increased risk for postoperative ventilation as well as a prolonged time to discharge [23]. However, a restrictive blood transfusion approach is only one pillar of patient blood management strategies, which additionally include a comprehensive management of anemia and a minimization of perioperative blood loss [24,25].

The choice of the anesthetic agent used for maintenance of anesthesia has also been demonstrated to be able to potentially influence outcomes after cancer surgery. In a large retrospective study analyzing data from more than 11,000 patients, the use of a propofol-based TIVA regimen significantly improved survival rates after cancer surgery compared to patients who had received inhalational anesthetics [26]. However, in the current study, no difference between TIVA and a balanced anesthesia using volatile anesthetics could be detected.

Additionally, neither the length of the ICU stay nor the time to hospital discharge were affected by intraoperative administration of PRBCs. However, the use of RA did have a considerable effect on these parameters: it was not only associated with fewer ICU admissions but also with a reduction of the (postoperative) LOS by three days. An LOS reduction is desirable for several reasons: it might reduce the risk of nosocomial infections and other complications [27], prevent crowded wards and ICUs, which is known to be associated with poorer outcomes [28,29], and might improve the hospital’s socioeconomic situation and patient satisfaction by allowing faster patient turnover [30]. A possible explanation for the observed decrease in LOS by RA might be due to its clinical benefits: less opioid consumption, earlier mobilization and ambulation, improved pulmonary function and overall fewer postoperative complications [31].

There have been numerous reports on a potential beneficial effect of LA and RA on outcomes and metastasis in patients undergoing cancer surgery, most of them being of retrospective nature [11,32,33]. Therefore, a Cochrane analysis from 2014 has rated the evidence to support a benefit of RA regarding tumor recurrence as being inadequate due to the lack of prospective, sufficiently powered clinical trials [34]. Several more recent meta-analyses also failed to provide sufficient evidence to prove the hypothesis of a possible anti-metastatic effect of the perioperative use of RA, although the quality of evidence might still be considered to be low [35,36,37]. However, several authors have provided experimental in vitro and in vivo data to support the hypothesis that LA might bear the potential to inhibit crucial signaling events in malignant cells, ultimately inhibiting the cells’ ability to metastasize [13,15,38,39,40]. The observed inhibition of signaling in malignant cells by LA is likely due to the strong anti-inflammatory properties of the drugs [14,41]. In the current study, perioperative use of RA did not affect OS and RFS.

However, a significant reduction in postoperative NLR was observed, which also might be related to the anti-inflammatory properties of LA and therefore be independent of the observation regarding the effects of intraoperative blood transfusion.

NLR has gained a lot of attention in recent years, as it could be identified as a prognostic marker for outcome in patients with cancer [42,43] and serves as a marker to assess and benchmark the patient’s systemic inflammatory response, for example after tumor surgery [44]. A higher NLR might be associated with worsened outcomes or lower response rates to treatments such as chemotherapy [45,46]. A reduced postoperative NLR, as seen in this study, might therefore suggest another beneficial aspect of the perioperative use of RA in cancer patients. However, the exact relevance of this reduced NLR remains unclear, as the analysis used in the current study was focused on the detection of a possible effect of the use of perioperative interventions on the NLR and not on its use as a prognostic factor for the course of the disease. In the current study, the use of RA decreased the postoperative NLR from 4.4 to 2.3. Unfortunately, there is not enough data available—especially not in pediatric cancer patients—to suggest a universal optimal cutoff value for the NLR in order to use it as a prognostic tool [47,48].

Obviously, the current study has several limitations. Due to its retrospective character, all results should be interpreted with caution. Although the multicentric design has significantly improved the power of the data, there was no specific study protocol to be followed and most perioperative decisions were left to the discretion of the anesthesiologist-in-charge. However, triggers for blood transfusion, for example, are strictly regulated by national guidelines, which all participating centers should have adhered to. Unfortunately, blood loss has not been recorded on a regular basis in all centers and can therefore not be analyzed. The longer duration of surgery in the PRBC group certainly suggests a more difficult surgical procedure in some cases, which might in turn also lead to a higher incidence of substantial blood loss requiring transfusion of blood products. Additionally, no information has been recorded regarding postoperative blood transfusions or during the further course of the disease.

Overall survival was over 90% in our study. With the rather small population of 65 patients in the current study and only a few events, i.e., deaths or relapses, the results of the Kaplan-Meier estimator and especially those of the Cox regression analysis should be interpreted with caution due to the high risk of bias [49]. In addition, the 95% confidence intervals for the hazard ratios observed in the Kaplan-Meier estimators as well as in the Cox regression analysis (see Table 4 and Table 5) were found to be large, suggesting a rather weak statistical association.

Another limitation might be the fact that differential blood counts for the determination of postoperative NLRs have not been conducted in all patients or on the same postoperative day. This extension of the postoperative period had been necessary to ensure enough valid data points for the analysis, but of course we are therefore also not able to rule out the possibility that this rather large range might have affected the presented observations. A cautious interpretation of the results is therefore warranted. However, the distribution of timepoints for the determination of the postoperative NLR did not differ significantly between the two groups. Despite the outlined limitations, which are mostly due to the retrospective nature of the study, however, a careful interpretation of the data as outlined above might still be justified.

## 5. Conclusions

To our knowledge this is the first study examining a potential association between anesthesia technique and outcome after pediatric cancer surgery. The results of this retrospective multicenter study support the hypothesis that anesthesia-related interventions during the perioperative period and the choice of the anesthetic regimen might play an important role for the outcome after such cancer surgery also in pediatric patients.

First, intraoperative blood transfusion significantly decreased RFS, suggesting a more restrictive blood transfusion regimen might be beneficial in these patients.

Second, the use of RA not only reduced ICU admissions and the LOS after surgery, but also decreased NLR postoperatively, providing more evidence for a significant anti-inflammatory effect of LA in patients with cancer.

Of course, these two novel findings should be interpreted separately, since the underlying mechanisms might differ. However, the study was conceived as a retrospective and exploratory study involving three large academic centers. Even with the appropriate caution regarding the interpretation of the results, the reported observations might still serve as excellent examples for hypothesis-generating and encouraging results, which might be worth exploring in a randomized clinical trial suitable for multivariate analyses in order to provide the best care possible to children suffering from nephroblastoma.

## Figures and Tables

**Figure 1 cancers-14-05585-f001:**
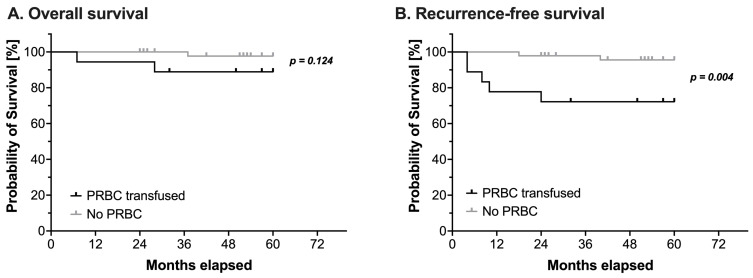
Kaplan-Meier estimates of (**A**) overall survival and (**B**) recurrence-free survival in pediatric patients undergoing surgery for nephroblastoma (*n* = 65). Survival analysis using a log-rank test compared patients with (black line) or without (grey line) intraoperative administration of packed red blood cells (PRBC). Censored events are marked with upright ticks. *P*-values as annotated.

**Figure 2 cancers-14-05585-f002:**
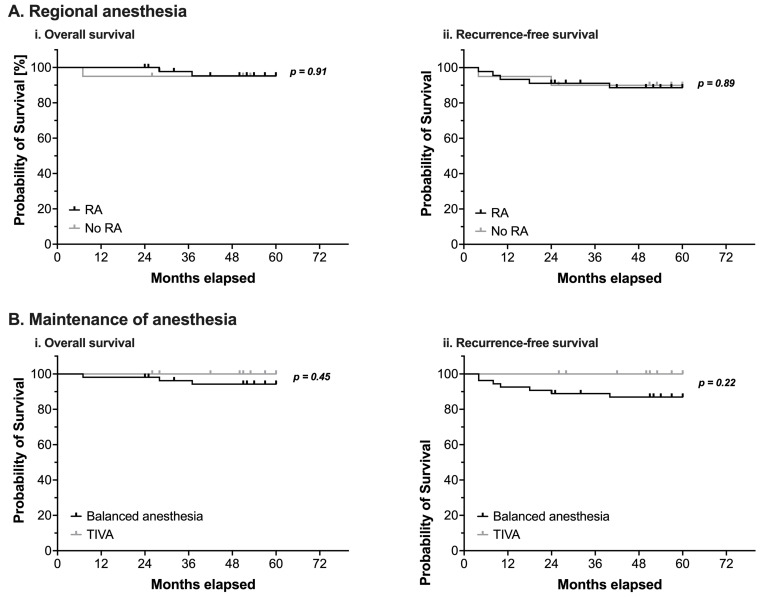
(**A**) Kaplan-Meier estimates of overall survival (**i**) and recurrence-free survival (**ii**) in pediatric patients undergoing surgery for nephroblastoma (*n* = 65). Survival analysis using a log-rank test compared patients with (black line) or without (grey line) intraoperative use of regional anesthesia (RA). (**B**) Kaplan-Meier estimates of overall survival (**i**) and recurrence-free survival (**ii**) in pediatric patients undergoing surgery for nephroblastoma (*n* = 65). Survival analysis using a log-rank test compared patients in whom anesthesia was either maintained with volatile anesthetics (balanced anesthesia, black line) or with total intravenous anesthesia using propofol (TIVA, grey line). Censored events are marked with upright ticks. *p*-values as annotated.

**Figure 3 cancers-14-05585-f003:**
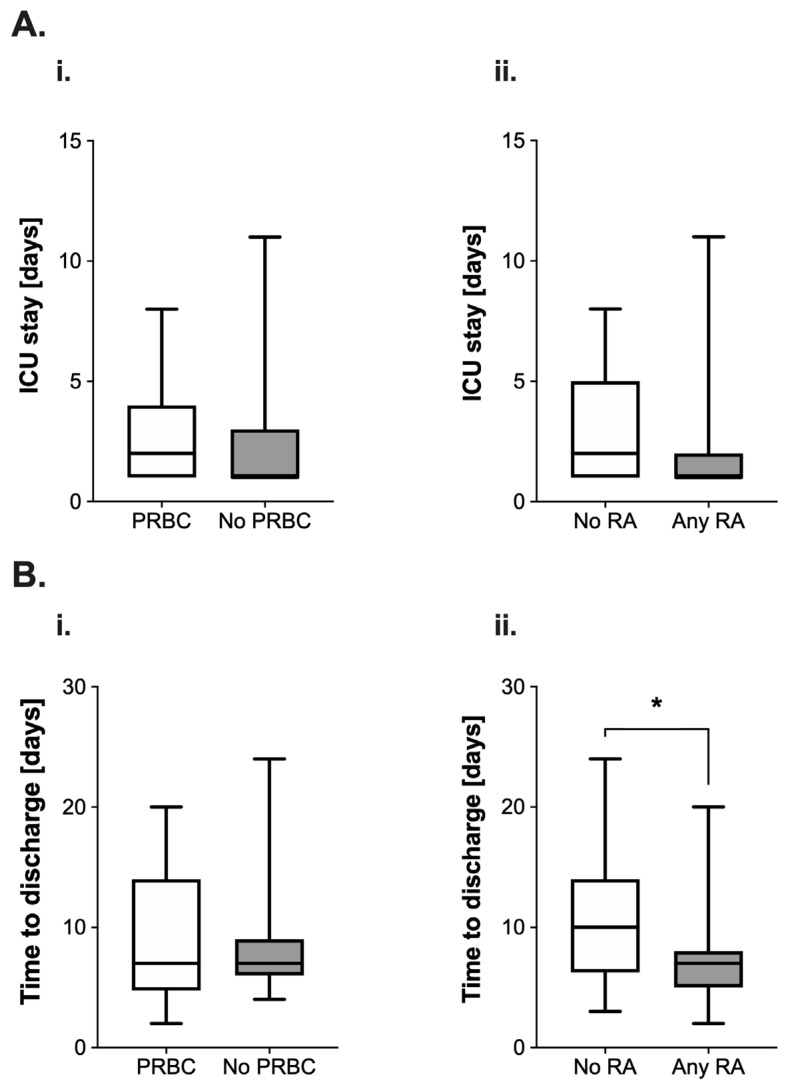
(**A**) Analysis of postoperative length of stay (in days) on the intensive care unit (ICU) in pediatric patients undergoing surgery for nephroblastoma. Data are presented as boxplots (median with quartiles, whiskers representing minimum and maximum). Patients were categorized into groups in accordance with their respective intraoperative treatment: (**i**) intraoperative administration of packed red blood cells (PRBC, white boxplot) vs. no administration of PRBC (grey boxplot) and (**ii**) no regional anesthesia as part of the anesthetic regimen (white boxplot) vs. perioperative use of regional anesthesia (RA, grey boxplot). (**B**) Analysis of the time to discharge (in days) after tumor surgery for nephroblastoma in pediatric patients. Patients were categorized into groups in accordance with their respective intraoperative treatment: (**i**) intraoperative administration of packed red blood cells (PRBC, white boxplot) vs. no administration of PRBC (grey boxplot) and (**ii**) no regional anesthesia as part of the anesthetic regimen (white boxplot) vs. perioperative use of regional anesthesia (RA, grey boxplot). * *p* < 0.05 for comparison as indicated.

**Figure 4 cancers-14-05585-f004:**
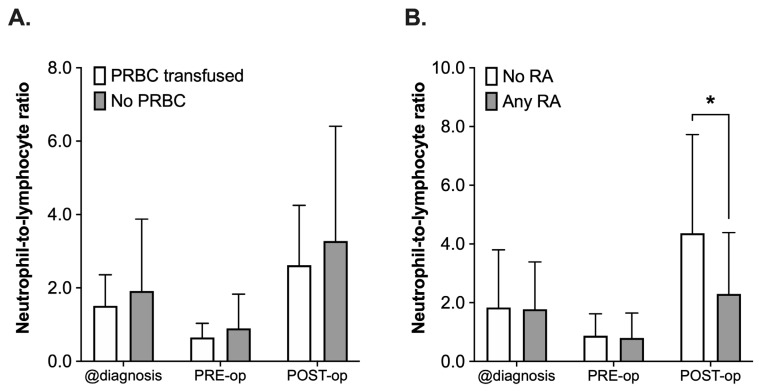
Analysis of neutrophil-to-lymphocyte ratio (NLR) in pediatric patients undergoing surgery for nephroblastoma at three different timepoints: at the time of diagnosis (@diagnosis), pre-operatively (PRE-op) and postoperatively (POST-op). Patients were categorized into groups in accordance with the according to their respective intraoperative treatment: (**A**) intraoperative administration of packed red blood cells (PRBC, white boxplots) vs. no administration of PRBC (grey boxplots) and (**B**) no regional anesthesia as part of the anesthetic regimen (white boxplots) vs. perioperative use of regional anesthesia (RA, grey boxplots). Data are presented as boxplots (median with quartiles, whiskers representing minimum and maximum). * *p* < 0.05 for comparison as indicated.

**Table 1 cancers-14-05585-t001:** Baseline characteristics of all patients. SD = standard deviation; ASA = American Society of Anesthesiology; IQR = interquartile range; RSI = rapid sequence induction; TIVA = total intravenous anesthesia; RA = regional anesthesia, PACU = post-anesthesia care unit, ICU = intensive care unit.

Parameter	All (*n* = 65)
Age (years; mean (SD))	3.8 ± 3.1
Weight (kg; mean (SD))	15.8 ± 9.2
Male (*n* (%))	31 (48%)
ASA classification	
I	0 (0%)
II	39 (60%)
III	25 (38%)
IV	1 (2%)
Tumor staging (*n* (%))	
I	32 (52%)
II	9 (15%)
III	11 (18%)
IV	8 (13%)
V	1 (1%)
Metastasis at diagnosis (*n* (%))	7 (11%)
Neoadjuvant chemotherapy (*n* (%))	57 (88%)
Duration of surgery (minutes; median (IQR))	204 (129)
Surgical approach	
Laparotomy (*n* (%))	64 (98%)
Laparoscopy (*n* (%))	1 (2%)
Duration of anesthesia (minutes; median (IQR))	315 (149)
Type of induction of anesthesia	
intravenous (*n* (%))	50 (77%)
inhalational (*n* (%))	4 (6%)
intravenous RSI (*n* (%))	11 (17%)
Maintenance of anesthesia	
balanced anesthesia (*n* (%))	54 (83%)
TIVA (*n* (%))	11 (17%)
Crystalloids (mL/kg BW; median (IQR))	63 (43)
Crystalloids (mL/kgBW/h; median (IQR))	13 (7)
Colloids used (*n* (%))	36 (55%)
Colloids (mL/kg BW; median (IQR))	20 (21)
Regional anesthesia	
none (*n* (%))	20 (31%)
any (*n* (%))	45 (69%)
Type of RA	
caudal block (*n* (%))	10 (15%)
epidural anesthesia (*n* (%))	35 (54%)
Postoperative care	
PACU/recovery room (*n* (%))	29 (45%)
ICU (*n* (%))	36 (55%)

**Table 2 cancers-14-05585-t002:** Baseline patient characteristics and perioperative data: Intraoperative blood transfusion. PRBC = packed red blood cells, SD = standard deviation; ASA = American Society of Anesthesiology; IQR = interquartile range; TIVA = total intravenous anesthesia; RA = regional anesthesia, PACU = post-anesthesia care unit, ICU = intensive care unit; Statistical tests as indicated: * Fishers Exact test; ° Students’ *t*-test; & Mann-Whitney-U test; § Chi-square test. Statistically significant results are shown in bold numbers.

Parameter	PRBC Transfused (*n* = 18)	No PRBC (*p* = 47)	*p*-Value
Age (years; mean ± SD)	4.4 ± 3.0	3.6 ± 3.1	0.37 °
Weight (kg; mean ± SD)	15.2 ± 9.4	17.2 ± 8.7	0.44 °
Male patients (*n* (%))	8 (44%)	23 (49%)	0.67 *
ASA classification			0.1 *
I	0 (0%)	0 (0%)	
II	8 (44%)	31 (66%)	
III	9 (50%)	16 (34%)	
IV	1 (6%)	0 (0%)	
Tumor staging (*n* (%))			**0.005 ***
I	5 (28%)	27 (63%)	
II	4 (22%)	5 (12%)	
III	7 (39%)	4 (9%)	
IV	1 (6%)	7 (16%)	
V	1 (6%)	0 (0%)	
Metastasis at diagnosis (*n* (%))	1 (6%)	6 (13%)	0.663 *
Neoadjuvant chemotherapy (*n* (%))	16 (89%)	41 (87%)	1.0 *
Duration of surgery (minutes; median (IQR))	271 (106)	171 (66)	**0.001 &**
Duration of anesthesia (minutes; median (IQR))	376 (123)	278 (76)	**<0.001 &**
Maintenance of anesthesia			0.713 *
balanced anesthesia (*n* (%))	16 (89%)	38 (81%)	
TIVA (*n* (%))	2 (11%)	9 (19%)	
Crystalloids (mL/kg BW; median (IQR))	80 (47)	67 (41)	0.005 &
Crystalloids (mL/kgBW/h; median (IQR))	13 (9)	14 (6)	0.403 &
Colloids used (*n* (%))	13 (72%)	23 (49%)	0.091 §
Colloids (mL/kg BW; median (IQR))	20 (29)	22 (22)	0.922 &
Regional anesthesia			
none (*n* (%))	6 (33%)	14 (30%)	0.782 §
any (*n* (%))	12 (67%)	33 (70%)	
Type of RA			
caudal block (*n* (%))	1 (6%)	9 (19%)	0.442 *
epidural anesthesia (*n* (%))	11 (61%)	24 (51%)	
Postoperative care			0.272 §
PACU/recovery room (*n* (%))	10 (56%)	19 (40%)	
ICU (*n* (%))	8 (44%)	28 (59%)	

**Table 3 cancers-14-05585-t003:** Baseline patient characteristics and perioperative data: Regional anesthesia. RA = regional anesthesia, ASA = American Society of Anesthesiology; IQR = interquartile range; TIVA = total intravenous anesthesia; PACU = post-anesthesia care unit, ICU = intensive care unit; Statistical tests as indicated: * Fishers Exact test; & Mann-Whitney-U test; § Chi-square test. Statistically significant results are shown in bold numbers.

Parameter	RA (*n* = 45)	No RA (*n* = 20)	*p*-Value
Age (years; median (IQR))	3.1 (3.6)	1.9 (3.4)	0.744 &
Weight (kg; median (IQR))	15.6 (8.5)	10.8 (7.7)	0.191 &
Male patients (*n* (%))	21 (47%)	10 (50%)	1.0 §
ASA classification			**0.025 ***
I	0 (0%)	0 (0%)	
II	31 (69%)	8 (40%)	
III	14 (31%)	11 (55%)	
IV	0 (0%)	1 (5%)	
Tumor staging (*n* (%))			**0.029 ***
I	19 (46%)	13 (65%)	
II	9 (22%)	0 (0%)	
III	6 (14.6%)	5 (25%)	
IV	7 (17.1%)	1 (5%)	
V	0 (0%)	1 (5%)	
Metastasis at diagnosis (*n* (%))	1 (5%)	6 (13%)	0.423 *
Neoadjuvant chemotherapy (*n* (%))	40 (89%)	17 (85%)	0.693 §
Duration of surgery (minutes; median (IQR))	187 (80)	274 (175)	0.201 &
Duration of anesthesia (minutes; median (IQR))	315 (72)	326 (150)	0.68 &
Maintenance of anesthesia			0.079 *
balanced anesthesia (*n* (%))	40 (89%)	14 (70%)	
TIVA (*n* (%))	5 (11%)	6 (30%)	
Crystalloids (mL/kg BW; median (IQR))	76 (35)	69 (58)	0.348 &
Crystalloids (mL/kgBW/h; median (IQR))	14 (6)	13 (12)	0.46 &
Colloids used (*n* (%))	27 (60%)	9 (45%)	0.291 §
Colloids (mL/kg BW; median (IQR))	23 (16)	10 (24)	**0.04 &**
Postoperative care			**0.001 ***
PACU/recovery room (*n* (%))	31 (69%)	5 (25%)	
ICU (*n* (%))	14 (31%)	15 (75%)	

**Table 4 cancers-14-05585-t004:** Hazard ratios. OS = overall survival; RFS = recurrence-free survival; PRBC = packed red blood cells; RA = regional anesthesia; TIVA = total intravenous anesthesia; HR = hazard ratio; CI = confidence interval. Statistically significant results are shown in bold numbers.

Type of Survival	Comparison	HR	95% CI	*p*-Value
**OS**	PRBC vs. No PRBC	5.37	0.42–68.4	0.124
	No RA vs. Any RA	1.14	0.1–13.37	0.91
	Balanced anesthesia vs. TIVA	3.3	0.15–70.8	0.45
**RFS**	PRBC vs. No PRBC	7.59	1.36–42.2	**0.004**
	No RA vs. Any RA	0.89	0.18–4.43	0.89
	Balanced anesthesia vs. TIVA	3.38	0.48–23.9	0.22

**Table 5 cancers-14-05585-t005:** Cox regression analysis. PRBC = packed red blood cells; RA = regional anesthesia; HR = hazard ratio; CI = confidence interval. Statistically significant results are shown in bold numbers.

Variable	Beta	SE	HR	95% CI	*p*-Value
Stage [I + II vs. III + IV + V]	0.73	0.78	2.07	0.45–9.53	0.351
PRBC [PRBC vs. No PRBC]	1.79	0.85	5.99	1.12–31.96	**0.04**

## Data Availability

The data presented in this study are available upon reasonable request from the corresponding author.

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
