# Peer review of "Association between Intraoperative Blood Transfusion, Regional Anesthesia and Outcome after Pediatric Tumor Surgery for Nephroblastoma"

_cancers, 2022, doi:10.3390/cancers14225585_

Round 1

Reviewer 1 Report

Dear authors, 

The ideea of your study is interesting and it has been the center of research in the field of regional versus general anaesthesia for some time. However, there are some very important limitations to the research. 

1. First of all, there is no power sample analysis and thus, taking into account the small number of patients, the results may be biased. Similar studies, moreover when taking into account outcome benefits, need to include a significantly higher number of patients in order to reach a power of at least 75%. 

2. There are some methodological aspects that need to be addressed:

- why include patients who already have metastases and those who have (or have not) undergone chemotherapy? this makes the group of patients uneven and is a major bias when considering outcome and cancer free survival (taking them out would make the cohort even smaller)

- patients underwent different types of anaesthesia: volatile vs TIVA. many studies have also focused on the potential benefits of this two on cancer free survival. This may also be a major bias. 

- please take a look at the type of surgery: laparoscopy vs laparotomy that may also influence outcome and length of stay. 

- NLR varies with inflammation and that varies with the time it has been measured. Since the timepoint of NLR calculation is not uniform (day 0 to day 8) the difference may be mostly due to this issue

3. Many randomized control studies have been conducted to investigate the potential benefits of RA on cancer reoccurence, including a recently published meta-analysis and failed to demonstrate any benefits. This should be cited and in this matter all the bibliography updated as most of the cited papers are quite old.

Overall, we thing that all of these should be addressed. 

Reviewer 2 Report

Thank you for submitting the manuscript. I read your study with great interest and in depth. The manuscript is well written, with a wealth of analysis and starts from an idea that is potentially interesting. However, I am extremely perplexed about the quality of your research. As you stated in the introduction, the evidence on the adult population is all derived from retrospective studies. Well, this level of evidence is very low, as it is neither prospective studies nor RCTs. If we then translate this evidence from the adult population (which certainly gets sick from other types of cancer than the child), the evidence fades even more. In addition, you also design a study as retrospective and not prospective, severely limiting the evidence. There is also considerable confusion as the primary outcome by definition is and must be only one. Instead, you evaluate both the effect of locoregional anesthesia and the effect of transfusions. This is a serious methodological error.

With regard to blood, much has been done in the pediatric field to reduce the indiscriminate use of transfusions and the results achieved and consolidated are of a good level. In fact, transfusions should be limited to those patients who need them due to comorbidities or symptoms, not only for a mere laboratory value. It is therefore completely uncertain to want to influence the decision to transfuse regarding a possible relapse on the neoplastic disease, when the patient would need it immediately to safeguard the myocardium, for example. I am therefore forced to suggest that you rethink the entire study, choosing the outcome well and thinking about an RCT. I hope these comments are helpful to you.

Reviewer 3 Report

The topic is very interesting although a major limitation which is not mentioned in the discussion is demonstrated in Table 2 - there is a clear trend in higher ASA classifications among the transfused patients and significantly higher stages of the tumor among the transfused group relative to the non-transfused group.  Also, there are twice the number of non-transfused patients as transfused patients - understanding that this is a retrospective study it would be recommended to increase the number of patients by increasing the number of academic centers involved.

Reviewer 4 Report

Dear Editor and Author,

This is a very good study. It has been known for some time that blood transfusion in cancer patients is unfavorable,

it causes immunization, which impairs the defense mechanisms against cancer.

This is especially important in the case of pediatric tumors. In my opinion, whole paper does not raise any substantive doubts. It is written in clear

and transparent scientific language. All abbreviations are clearly explained.

The methodology, results and conclusions are detailed. Nevertheless, I fully recommend accepting the paper for publishing in Cancers. Regards

Round 2

Reviewer 1 Report

 This is an interesting study evaluating the potential impact of intraoperative administration of erythrocyte concentrates, the use of regional anesthesia, and the choice of the anesthetic regimen on overall and recurrence-free survival in children with nephroblastoma. The authors found that intraoperative administration of erythrocyte concentrates was associated with a reduction in recurrence-free survival, while the use of regional anesthesia and the choice of anesthetic used for maintenance of anesthesia did not demonstrate an effect on the primary outcomes.

In the present version of the manuscript, the authors already made  the changes suggested by the reviewers and addressed the reviewers’ questions and concerns adequately.

The topic of the study- the influence of blood transfusions and of anesthetic techniques on the recurrence of cancer is extremely interesting and there are many publications that appeared in this field. However, most published research is performed in adults, while in children the data is very sparse. The present research has indeed limitations, as it is retrospective and performed on a relatively small number of patients but the authors acknowledge these limitations and draw attention that data obtained here should not lead to over-interpretation. However, this is valuable research that can serve as a hypothesis-generating study, and more research (preferably prospective and ideally RCT) should follow.

The manuscript is well-written, easy to read, and for sure interesting for many readers of the journal. 

Reviewer 2 Report

I have reviewed the corrections but I think that the result is not adequate for publication.

Reviewer 3 Report

I greatly appreciate the addition of Table 5 - I would add an abbreviated portion of your comments to the discussion and I would then recommend publication.  Thank you
